



# Barriers to mainstream adoption of catchment wide Natural Flood Management, a transdisciplinary problem framing study of delivery practice

Thea Wingfield[1], Neil Macdonald[1] Kim Peters[2,3,4] and Jack Spees[5]

(1) University of Liverpool, School of Environmental Sciences,

(2) Alfred-Wegener-Institut Helmholtz-Zentrum für Polar- und Meeresforschung, Am Handelshafen 12, 27570 Bremerhaven

(3) Helmholtz Institute for Functional Marine Biodiversity at the University of Oldenburg (HIFMB), Ammerländer Heerstraße 231, 26129 Oldenburg

(4) Institute for Chemistry and Biology of Marine Environments [ICBM], University Oldenburg, Carl-von-Ossietzky-Straße 911, 26133 Oldenburg

(5) The Ribble Rivers Trust, Clitheroe, Lancashire, United Kingdom

*Correspondence to*: Thea A.J Wingfield (t.a.j.wingfield@liverpool.ac.uk)

**Abstract.** Natural Flood Management (NFM) is the name given to Nature Based Solutions (NBS) for Flood Management in the UK. It is a holistic flood management technique that employs natural hydrological processes, through the instillation of interventions, to slow the flow of water, creating a landscape scale flood management system. Despite widespread interest and supporting policy from governments and non-profit organisations NFM, as yet, has not been widely adopted as a mainstream flood management technique. A small number of academic studies examining perceived barriers to NFM adoption have identified a variety of individual factors as being responsible. It is commonly accepted that flood risk management broadly, and NFM specifically, are complex, challenges of interacting physical and human parameters and that academic, institutional and policy divisions are rarely sympathetic to embracing these complexities. A transdisciplinary problem framing study in conjunction with professionals experienced in the delivery of NFM projects in the UK aimed to capture these multifaceted parameters of flood management and strategic delivery at a landscape scale using Group Concept Mapping, a systems approach to identify conceptual convergence. This policy-delivery impasse was further explored by quantifying the relative importance of individual barriers and conceptual groupings from the perspective of two different practitioner groups (flood risk managers and conservation practitioners). The results demonstrate that the NFM delivery system can be grouped into seven interacting elements: *policy and regulation*, *politics*, *public perception*, *cross-cutting issues*, *funding*, *technical knowledge and evidence*, of which each have a varying number of barriers that limit NFM uptake. Opinions differs as to the importance of these individual barriers, however when considering the system broadly we identify that the institutional and social barriers are perceived as the most important, whilst technical knowledge and evidence are the areas of least concern. This paper aims to promote NBS flood management delivery in the UK and globally by generating, structuring and representing the multifaceted and multilevel NFM delivery system at a local level to evidence adaptive decision making at a regional, national and global



level. Through problem structuring and an increased understanding and awareness of the structure and network of linking elements and perceived differences of practitioner groups that influence the system of delivery, steps can be taken towards

solutions that are socially, scientifically and practically robust.

## 1 Introduction

In the UK Natural Flood Management (NFM) is the commonly used term for Nature Based Solutions (NBS) for flood management; a holistic flood management technique designed to mimic natural environmental conditions by harnessing

hydrological processes to slow water flowing through the landscape (Werritty, 2006). The terms NFM and NBS for flood management (UN-Water, 2018) indicate flood management as the primary goal of the techniques. In practice,  proponents of their use emphasise their transformative strength in the delivery of multiple benefits (Barlow et al., 2014; Forbes et al., 2015; Hanson et al., 2020). Promising flood risk reduction with greater environmental and social goods (Connelly et al., 2020; Fenner, 2017), including improved biodiversity (Cook et al., 2016), water quality (Barber and Quinn, 2012; Howe and White,

2003), public health and well-being (de Bell et al., 2017; Bratman et al., 2019; Maas, 2006) and carbon sequestration (WWAP 2018). NFM describes methods that restore (Lane, 2017) or mimic (Barber and Quinn, 2012) hydrological processes within the water cycle, including engineered land forms (Wingfield *et al*., 2019). The pinnacle of the practice is to link a large number of small interventions through connectivity (Keesstra et al., 2018) to optimise a system functioning at a landscape scale delivering a cumulative catchment wide land and water management strategy. Sectors needed to deliver coordinated activities,

at multiple scales, for strategic delivery include urban planning and development, agriculture and conservation, flood and coastal risk management and water and waste water supply and management. At present the activities and legal frameworks of these sectors in the UK are not aligned (Goytia, 2021), nor does NFM have a well-defined delivery program with a clear champion to drive real action and/or empower others to change (Wingfield *et al.* 2019). To date NFM has not been widely adopted as a mainstream flood management technique (Bark et al., 2021) with some practitioners continuing to regard it as a

novel approach (O'Donnell et al., 2017; Schanze, 2017).

Techniques of restoring and mimicking hydrological processes are employed across the globe but communication across sectors, disciplines and locations is fragmented under different terminology (Fletcher et al., 2015) including; blue-green infrastructure (BGI) (Gaffin et al., 2012; Stovin and Ashley, 2019), water sensitive urban design (WSUD) (Kuller et al., 2017),

the sponge city (Liang et al., 2020), ecosystem based adaptation (Faivre et al., 2018; Rondón-Krummheuer et al., 2015), sustainable drainage (Jones and Macdonald, 2007), low impact development (LID) (EPA, 2000), natural flood management (NFM) (Lane, 2017) and nature based solutions (NBS) (Hartmann et al., 2019; Keesstra et al., 2018). Practitioners and researchers regularly encounter new terminology, often loosely defined and conceptually broadly similar leading to complaints of an excess of green jargon (Nesshöver et al., 2017). Whether NBS will mature beyond 'just another communication tool', to





playing a pivotal role in land and water management will depend on generating new knowledge about how to implement them across 'scales, contexts and people' (Albert et al., 2017; Schanze, 2017). Venkataramanan *et al.* (2020) amongst others echo this conclusion stating that 'ample evidence' exists supporting technical efficacy whilst knowledge is lacking in understanding the socio-ecological-technical system of delivery. (Carlet, 2015; Dhakal and Chevalier, 2017; Li et al., 2019; Nesshöver et al., 2017; Venkataramanan et al., 2020).


Research is required to reveal transformative pathways for the incorporation of NFM into a mainstream flood management strategy that conceptualises NFM as dynamically complex, which can reveal associations between social and environmental domains (Gómez Martín et al., 2020; Keesstra et al., 2018). Within social-ecological systems multiple actors bring differing perspectives of the nature of problem(s), different interpretations of potential solutions and ambiguity as to who is responsible

for applying and operating any new method. Academic disciplines, and the career and journal systems that serve them, generate epistemological limitations creating barriers between actors in problem framing, application of methods and ideals of 'proof' (Hazard et al., 2020; Mauser et al., 2013). Furthermore a technocratic model of transmitting scientific fact without engaging in dialogue across disciplines and between science and society is seen in the academic literature's focus on NFM effectiveness as a measure of flood peak reduction or delay (Connelly et al., 2020; Dadson et al., 2017; Wingfield et al., 2019) is insufficient

for transformative decision making (Young et al., 2014). A growing number of voices concerned with sustainable futures have called for a change to scientific knowledge generation, to instead foster knowledge co-production with academic and non-academic actors to develop integrated research questions, services, policies and processes (Mauser et al., 2013; Pereira et al., 2018; Schneider et al., 2021). System oriented research also known as interdisciplinary or transdisciplinary research aims to 'transcend disciplinary and methodological paradigms' (Hadorn et al., 2008) to foster knowledge exchange and generate new

mental models and practices for sustainability oriented action (Pereira et al., 2018).

Environmental practitioners, work within an established system reinforced by embedded ways of working, governing and thinking about the system (Bark et al., 2021; Ngai et al., 2020; O'Donnell et al., 2018) The primary focus of NFM guidance written for practitioners is in supporting quantification of ecosystem processes and engineering principles to design

interventions (Burgess-Gamble et al., 2017; CIRIA, 2018; http://nwrm.eu/, 2013; The Environment Agency, 2017). Guidance documents are not explicitly addressing the interdisciplinary nature of NFM research, planning and delivery. Mainstream delivery of NFM requires a paradigm shift that incorporates social, environmental and ecological dimensions (Dekker and Fantini, 2020; Werritty, 2006), those practitioners with responsibilities for designing, applying for funding and/or delivering projects have valuable knowledge and experiences from navigating these complex dynamics.


This paper describes a methodological approach, and outputs of a problem framing component of a wider transdisciplinary study (2016 – 2020) between academics and practitioners in the UK with a shared interest in promoting the mainstream adoption of NFM. The authors are a doctoral research student who designed and led the project and a multi-discipline and





practitioner supervisory team that draws interdisciplinary expertise from flood research, human geography, governance, time
space and territory, catchment science, fundraising and partnership working.   The process of a transdisciplinary research
program begins with problem framing, knowledge co-production and integration of interested actors (Norström et al., 2020;
Pohl et al., 2021; Schneider et al., 2021). To do this, Group concept mapping (GCM) (Trochim 1989) with Ketso (Tippett et
al., 2007), a systems thinking methodology (Hassmiller Lich et al., 2017) that applies a mixed method, data collection approach
was used.  While not explicitly developed for use in transdisciplinary research, it is a useful tool, allowing for the emergence
of contested knowledge, differences in knowledge framing or highlighting blockages in knowledge transfer. The GCM method
produces visual representations of what a group is thinking on a particular topic (Donnelly and Ph, 2016) and in doing so
enables integrated problem identification, the primary component of transdisciplinary research (Jahn et al., 2012; Lang et al.,
2012; Pohl et al., 2021).

Rather than limit the study to merely identifying individual barriers to NFM delivery, our aim is to examine interdependencies
and identify conceptual convergence within the delivery system. In doing so the study reveals conditions in which barriers to
the delivery of NFM persist and begins to identify areas for further research and intervention points that could act as a catalyst
for change (Eisenack et al., 2014). Our aim in describing the methodological approach and sharing findings is to support NBS
adaptive decision making in contexts outside of North West England by revealing and examining  problems of this particular
system (Biesbroek et al., 2013). In-depth case studies, when used and reviewed alongside the natural science evidence base
can act as a catalyst to 'activate' hydrology research (Maruyama, 2001). The principle of connecting research practice and
theory with input from practitioners guided our approach and in doing so the outputs have the potential to be transferred to
other settings (Bickman et al., 2016).  It is recognised that while data collection is limited to England, questions on how to
implement NBS for flood management are of interest globally (Gómez Martín et al., 2020; Sowińska-Świerkosz and García,
2021). Contexts and actors involved differ across the globe but our systematic method that brings together diverse groups of
actors in a process of knowledge co-production is informative globally. Our approach can be replicated by multidisciplinary
groups interested in comparing, contrasting and evaluating patterns and structures within environmental management systems
to identify transformative pathways and knowledge gaps.

The paper begins by demonstrating a discrepancy between policy and delivery in relation to landscape scale holistic flood
management techniques and frames NFM delivery as a wicked problem requiring a research approach that transcends
disciplinary and methodological paradigms integrating knowledge of academic and non-academic actors. The paper then
proceeds to investigate how Group Concept Mapping with Ketso was used as a structured method to engage in dialogue and
co-create knowledge across disciplines and practitioner groups. The paper first explains an established approach of studying
barriers and adoption of sustainable management paradigms and then combines this approach with transdisciplinary problem
framing and a step-by-step reflection on using Group concept mapping with Ketso it in practice.  It does so to demonstrate its
potential and provide a framework for integrative systematic problem oriented research that creates visual representations of



networks of ideas to describe how a group thinks about an issue to be used in developing mutual understanding as a basis for further research and cooperation.


## 2. Barriers

Barriers are widely studied in reference to modern sustainable management paradigms, such as climate change adaptation (Biesbroek et al., 2013; Eisenack et al., 2014; Moser and Ekstrom, 2010) and sustainable agricultural practices (Bustamante et al., 2014; Rodriguez et al., 2009). Barriers are distinct from limiting factors in that, barriers within a system, once they are

identified can be managed and overcome, where as a limiting factor within that system is unsurmountable (Eisenack et al., 2014). Policy documents of environmental regulatory authorities in England encourage the use of NFM, describing it as: 'an option more resilient to extreme events' (Environment Agency, 2011), 'better value for money and a flexible and resilient solution that can deliver socioeconomic and environmental benefits alongside flood risk management benefits' (Penning-Rowsell, 2010). However, NFM as yet is not embedded as a mainstream flood management strategy (Bark et al., 2021), our

understanding on why remains somewhat fragmented, a number of explanations have been put forward (Table 1).

Table 1. Selection of published papers and reports identifying perceived barriers to NFM delivery

| Title | Date | Author(s) | Actors and location | Barriers to NFM delivery |
|---|---|---|---|---|
| Natural flood management from the farmer's perspective: criteria that affect uptake | 2017 | K.L. Holstead, *et.al* | Upland farmers. Scotland | 1. Lack of financial incentives for farmers<br>2. Limited information and education for farmers |
| Challenges to enabling and implementing Natural Flood Management in Scotland | 2017 | K. A. Waylen, *et.al* | Flood management organisations Scotland | 1. Lack of resources<br>2. Gaps in evidence base<br>3. Culture change required within flood risk management to collaborate more freely |
| A restatement of the natural science evidence concerning catchment-based 'natural' flood management in the UK | 2017 | S.J. Dadson, *et.al* | Informed flood risk management audience. Principally UK (fluvial flooding) | Whilst the review does not explicitly explore barriers to adoption it does conclude that there are uncertainties around quantitative predictions of flood risk reduction and co-benefits |



| Title | Date | Author(s) | Actors and location | Barriers to NFM delivery |
|---|---|---|---|---|
| Natural flood management | 2017 | S.N. Lane | Informed flood risk management audience. | 1. Scientific uncertainties at larger spatial scales <br> 2. NFM is a wicked problem |
| Natural Flood Management: Beyond the evidence debate | 2019 | T. Wingfield, *et.al* | 4 sectors (Flood risk management, conservation and agriculture, urban planning and development, water supply) UK | 1. Research limited to numerical modelling <br> 2. Misaligned policy objectives and activities of the four sectors involved in delivery |
| Barriers & solutions to mainstreaming Natural Flood Management within the Capital Programme | 2019 | Environment Agency report | Environment Agency staff England | 1. Culture change required within flood risk management to collaborate more freely <br> 2. Internal systems unsupportive of NFM <br> 3. Lack of clarity over maintenance and liability |
| Nature-based solutions for hydro-meteorological hazards: Revised concepts, classification schemes and databases | 2019 | Debele *et al.* | Systematic literature review of nature based solutions for hydro-meteorological hazards | 1. Gaps in technical knowledge <br> 2. Fragmented approaches and lack of collaboration amongst sectors |
| Stakeholders' views on natural flood management: Implications for the nature-based solutions paradigm shift? | 2021 | Bark *et al.* | UK water and environmental management stakeholders | 1. Lack of shared understanding of how to enable and implement NFM <br> 2. Private land owners with 'rights' over land use decisions |

The literature summarised in Table 1 demonstrates that investigations examining NFM delivery are focused principally on the
flood risk management sector, with only two of the studies bringing together perspectives of different stakeholder groups. Deficiencies in the evidence base is a theme common amongst papers from 2017, but from 2019 a shift towards mechanisms of delivery can be seen (Environment Agency, 2019; Wingfield et al., 2019). Waylan *et al* (2017), while highlighting a number



of barriers, including a lack of resources and a need for more evidence, strongly advocates a need for culture change within flood risk management itself. This finding is echoed in the internal review undertaken by the Environment Agency (2019).

Institutions responsible for flood management are resistant to change (Buuren et al., 2015; Sarabi et al., 2019), however NFM delivery by its very nature transforms the scale at which flooding and flood generation processes are considered, from local to catchment. It is largely because of this change in scale that strategic planning and delivery is needed amongst sectors that are not currently coordinating activities (Wingfield et al., 2019). To date no study has been undertaken at this scale, specifically designed to examine delivery practitioner interactions within which NFM delivery is situated, in a process of knowledge co-

creation, across disciplines and with academic and non-academic actors. This study aims to address this gap by targeting practitioners who contribute to catchment partnerships (Collins et al., 2020) and flood and coastal erosion risk management professionals.

## 3. Research approach and methodology

NFM delivery is influenced by social, technical and biophysical factors that interact and feedback in a multitude of complex ways, a characteristic commonly described as a 'wicked problem' (Lane, 2017; Rittel and Webber, 1973). Unlike "good" problems, which operate within defined rules (Brown et al., 2010), wicked problems are dynamic, and an approach limited to the methods and perspective of one disciplinary domain are insufficient. Research aimed at problem resolution, such as how to deliver catchment wide NFM, is required to address multifaceted socio-ecological problems involving interdisciplinary

research techniques and researchers to work with practitioners to understand and improve the system (McTaggart 1991). Benefits of researchers and practitioners working together is well established in the participatory research literature, indeed, this structure is reflected in the interdisciplinary project team, with one of the authors a director of a host organisation in North West England. The inclusion of a host organisation director as part of the project team (and as a PhD supervisor) offered insights and provided access to a complex system, but did not determine any research priorities, nor shaped or determined

research findings, as they were not present during workshops or data collection/analysis events. Throughout we were conscious of the positionality of the research team and potential implications on the research process and sought to minimise potential bias and influence, which was an additional benefit of the methodological approach applied. Novel findings and opportunity for change are generated by empowering those individuals who are most familiar with the system under scrutiny to steer the research agenda, data generation and application of findings (Newing et al., 2011). An interdisciplinary, participatory

approach, embraces socio-ecological complexity (Jantsch, 1972) rather than a traditional position, which tends to smooth out variation. Practitioners of NFM are a heterogeneous group, individuals will differ in their perspectives, ideas and interactions with other actors in response to different disciplinary backgrounds and a range of values and conceptual framings. Research that harnesses processes of knowledge co-production (Mauser et al., 2013; Pereira et al., 2018; Schneider et al., 2021) and social-learning to explore and clarify differences can reveal a range of values and framings (Keen et al., 2005; Pahl-Wostl et

al., 2007), whilst encouraging deconstruction of ineffectual old ideas and construct shared new possibilities. Conceptual



changes seed shifts in scientific and policy discourse that have been shown to bring about paradigm shifts (Bark et al., 2021; Pahl-Wostl et al., 2011).

### 3.1 Participant Identification

The integration of land and water management to harness hydrological processes to increase the hydrological resilience of the system is a messy real-world problem that has failed to move from a two decade-old policy ambition. A previous study identified that within the UK environmental governance system, catchment partnerships are well placed to co-ordinate delivery as the integrated water management framework that steers the movement is comparable and compatible to NFM in encouraging the delivery of multiple benefits coordinated at a catchment scale (Wingfield et al., 2019). Schanze gives an alternative view

(2017) calling for research that advances knowledge to support NFM delivery within flood risk management practice. This is the first study that brings together these two practitioner groups to examine these two perspectives in transdisciplinary and knowledge co-production research (Norström et al., 2020; Pohl et al., 2021; Schneider et al., 2021).

### 3.1.1 Flood Risk Authorities

In England, risk assessment management authorities (RMA's): The Environment Agency, lead local flood authorities (LLFA's), district and borough councils, coast protection authorities, water and sewerage companies, internal drainage boards and highways authorities, are responsible for the delivery of flood risk management policy. In 2020 the English government published a new national flood and coastal erosion risk management strategy that places greater emphasis, than previous strategies, on nature based solutions and using catchment based approaches (DEFRA, 2020). Current flood models do not

satisfactorily simulate hydrological processes of an NFM integrated system at a catchment scale (Metcalfe et al., 2017; Pattison and Lane, 2012), therefore it can be challenging to demonstrate the 'economic' value NFM delivers of flood risk reduction to properties and infrastructure. Furthermore costs of commissioning modelling studies to justify spending on an NFM scheme can exceed the costs of its delivery (Burgess and Hill, 2018) reinforcing spending skewed towards traditional, hard engineered flood defence system(s). Flood risk management responsibility in England is divided amongst RMAs according to sources of

flooding: river, coastal, groundwater, surface water, reservoir and sewerage. NFM and its use of hydrological processes to slow water in the landscape as a flood risk management technique employed to manage all types of flooding does not align with the delivery responsibilities of one RMA over another.

### 3.1.2 Catchment partnerships

Catchment partnerships were established in 2011 to deliver integrated water management by bringing together different sectors and organisations into a cooperative forum to facilitate greater integrated land and water management activities. They are led by a 'host' organisation, a charitable body, typically an Environmental non-Government Organisation (Collins et al., 2020) working at the local level (Department for Environment Food and Rural Affairs, 2015). Alongside awareness raising, typical of conservation organisations (Mace, 2014) the biological disciplinary framing of host organisations are evident in the majority



of their activities designed to enhance biodiversity, protect wildlife and restore habitats (CaBA, 2018). Partnerships use a ground-up model (Koontz 2004) to encourage interested parties to collaborate, identify mutual gains and deliver multiple benefits, rather than relying on a top-down regulatory approach to drive environmental improvement works. This has become known as the catchment based approach (CaBA, 2018). It is noted that the CaBA model is not yet proven (Watson, 2015), they face acute financial uncertainties, receiving limited centralised government funding and their formation is based on good will,

rather than through regulatory reform. Nevertheless a review of priority working areas of the 102 different catchment partnerships in England identified flood risk to be of interest for 93% of catchment partnerships (CaBA, 2017) and given that NFM is an integrated catchment management technique, catchment partnerships are well positioned to lead and influence NFM delivery (Collins et al., 2020).

**3.2 Group Concept mapping**

Group concept mapping (GCM) (Trochim 1989) with Ketso (Tippett et al., 2007) is a mixed method, data collection approach that visually represents the ideas of a group using multivariate quantitative analysis and creates opportunities for qualitative data interrogation.

The output of GCM with Ketso is described as a map, it displays previously unconnected ideas clustered into concepts that

reveal how and what a group and sub-groups think on a topic (Donnelly and Ph, 2016). Crucially, given the participatory nature of this study, it offers an opportunity for the participants to discuss and explore together the interconnected physical and human parameters of the system, incorporating differing perspectives of actors and the significance of diverse or fragmented conceptualisations (Cabrera, 2006; Tippett et al., 2007). Additionally a quantitative element of the approach allows for an empirical assessment of the perceptions and conceptualisations of a group, resulting in more robust data than achieved

through expert opinion alone.

GCM with Ketso is formed of six steps, which for the purposes of this study were grouped into 3 phases; phase 1 qualitative statement generation, phase 2 quantitative statement sorting and ranking and phase 3 interpretation (Figure 1).

Phase 1 and 2 were conducted between April 2016 and September 2016.



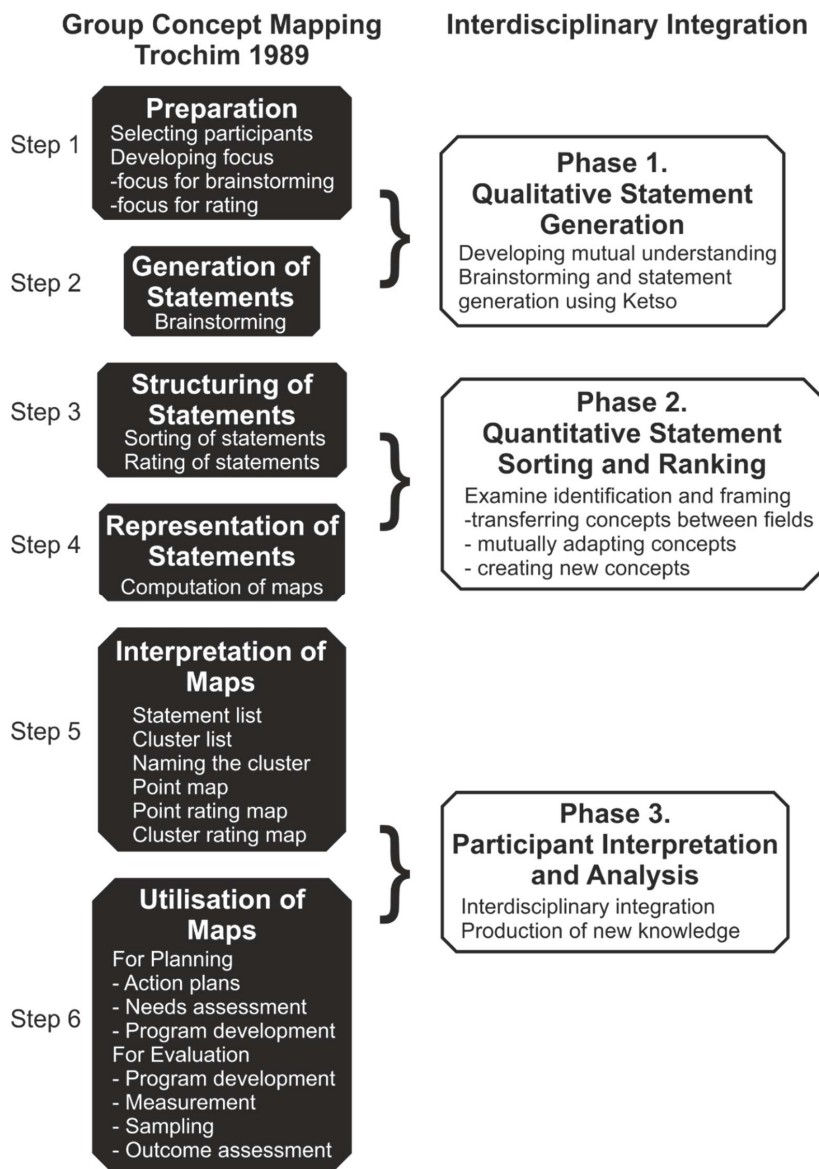

Figure 1. Group Conceptual Mapping (GCM) methodology with Ketso adapted from Trochim 1989 for interdisciplinary integration


### 3.2.1 Phase 1 qualitative statement generation

Phase 1 was completed in two sessions. The first through a workshop at the River Restoration Conference in Blackpool on the 27th April 2016. 39 of the conference attendees took part in the workshop enabling access to a heterogeneous practitioner group

drawn from across the UK, many of whom are expert in their fields including contractors, engineers, consultants, academics, environmental NGOs and government agencies (Wingfield, 2016). The second workshop was attended by 12 practitioners from the Environment Agency National Capital Programme Management Service (NCPMS), individuals responsible for delivering the Flood Risk Capital Programme. This team was selected as having the most comprehensive knowledge of flood risk management scheme delivery via their responsibility for managing the multi-million pound budget allocated to large flood

management schemes across England.

In both the large (n39) and small (n12) workshops participants were divided into groups of between six and eight in order to generate statements for phase 2 (Figure 1). The generation of statements creates a set of ideas that together capture all elements used to describe a system, values and/or concepts of the study domain. A number of techniques can be employed in a research setting to generate statements including use of a predetermined statement set, text abstraction and keywords, however

brainstorming is perhaps the most commonly used and familiar technique. Brainstorming is most suited to the combining of ideas and knowledge across disciplinary and operational divides as participants express themselves in their own words. This study used a form of live structured brainstorming called Ketso (Tippett and How 2011; McIntosh and Cockburn-Wootten 2016). Each group undertook a structured conversation using the Ketso methodology (Tippett and How 2011; McIntosh and Cockburn-Wootten 2016), a technique for structuring dialogue in a workshop environment. The advantage of Ketso is that it

is specifically designed to allow all participants to contribute without one or a few voices dominating, thereby mirroring the approach of GCM in representing the ideas of a group rather than expert opinion. Second, the technique employs visual, active and oral aids to support the developing discussion (Figure 2). The Ketso methodology can draw out ideas or themes and make connections that might not otherwise be identified using a traditional focus group discussion (Tippett and How 2011) or brainstorming activity.

A conversation builds in stages around a central question. In this case, 'how can the delivery of catchment wide NFM be encouraged?' The first element considered by each group was, 'what are the foundations of the NFM delivery system?' The second asked, 'what are the emerging mechanisms that encourage wider NFM delivery?' for example; actions, tools, objectives or organisations? The third generated data which forms the basis of this paper, 'what are the barriers to delivery?' And finally the last was 'what could be done to encourage and develop the emerging mechanisms or address any of the barriers.'

A formal definition of a barrier to NFM was not made for the study. The participants were left to interpret this for themselves to avoid consciously or sub-consciously influencing their understanding and therefore potentially, their view of challenges, restrictions or problems of NFM delivery. This subjectivity becomes data in itself and is explored in the second research question as to the differences in perceptions between practitioners.

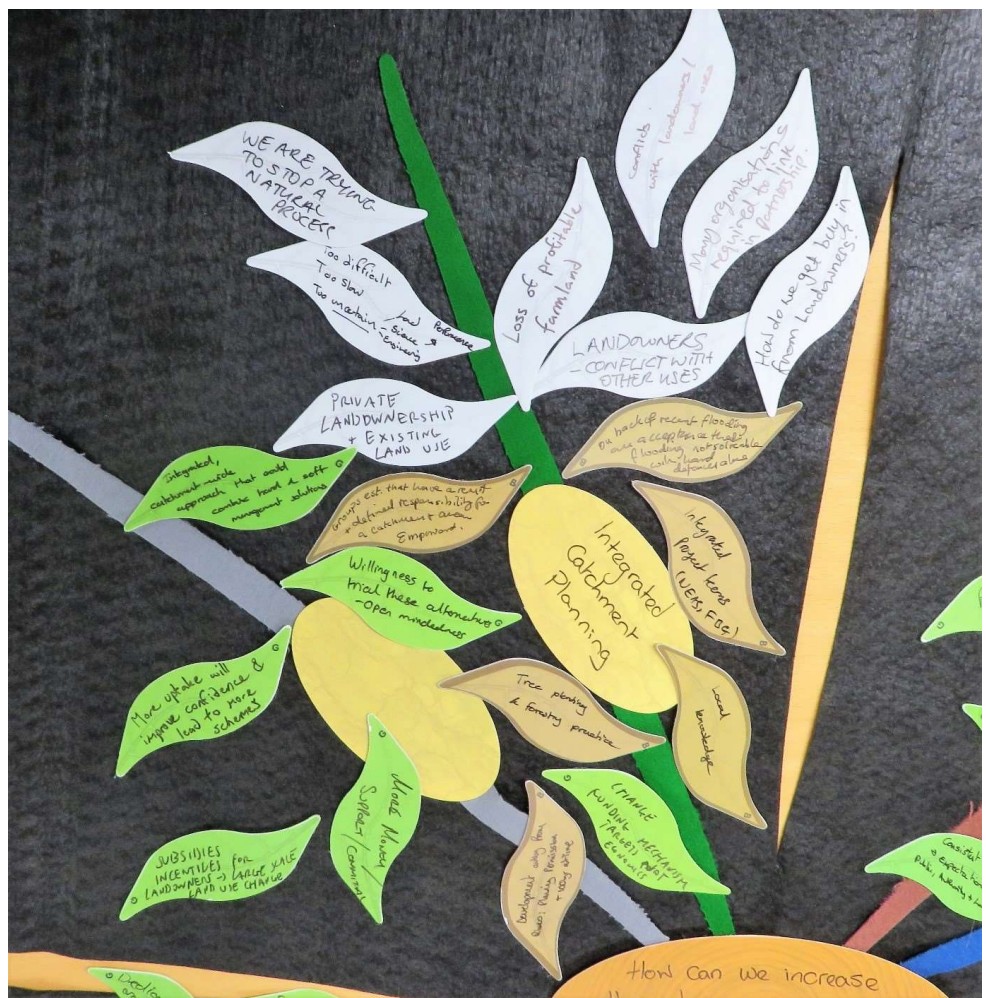

Figure 2. Example output from the Ketso structured conversation method examining the adoption of catchment wide NFM.

### 3.2.2 Phase 2 qualitative statement sorting and ranking

A total of 253 barrier statements were generated during phase 1, these were sorted and reduced to a final list of 47 for the
second phase of the group concept mapping methodology. The process had three purposes:

285        1) reduce the number of statements to a reasonable number for participants to manage in one session, whilst ensuring each
        barrier remains distinct to make comparisons;



2) check each statement for clarity and remove any deemed incomprehensible; and,

3) combine similar statements (Cabrera, 2006).

An independent reviewer was used to assist the primary researcher and review the validity of the initial statement reduction.

Qualitative statement sorting and ranking (phase 2, Figure 1) was then undertaken by 12 flood risk management professionals and 12 practitioners who contribute to catchment partnerships. The principal researcher provided guidance to the participants, either alone or in small groups. Whilst national experts were the target of phase 1, statement generation; practitioners who work within North West England were selected for the second phase. We were interested in testing the experiences and perceptions of the two different professional groups resulting from planning and delivering NFM interventions and therefore

wanted to reduce any influence of biophysical and socio-technical factors resulting from practice in different geographical regions that may complicate comparisons (Dupuis and Biesbroek, 2013). This two-stage process helps reduce the significance of local place or site within the study, however the national legislative context remains important in framing policy.

The exercise involved each participant sorting the statements into groups that they felt contained similar or related ideas and

classify the group by giving it a name. Participants were then asked to rank the importance of each individual statement, using a 0-7 likert scale. The scale was low to high, 1 indicating that the barrier is unimportant or disagreed with and a 7 indicating that the barrier is extremely important, 0 was included to enable practitioners who perceived they did not have sufficient knowledge to give an opinion.

All 24 participants sort and rank the statements (n47) after which this qualitative data is transformed through quantitative analysis (Step 4, Figure 1) to reveal the way in which the group, as a whole and the two practitioner sub-groups create meaning. A group similarity matrix (Trochim 1989) combines all 24 participant's outputs into a grid square with as many columns and rows as statements (n47), from which a concept map is drawn using multidimensional scaling (Hout *et al.* 2013, Kruskal, and Wish, 1978). Each statement is located on a grid square according to their similarity and dissimilarity. For this analysis a two-

dimensional solution locates points representing each barrier statement on an X-Y graph (Figure 3a). Ward's cluster analysis is applied to the multidimensional scaling outputs. Clustering statements into concept groups allows for many individual ideas, that could be overwhelmed, to be brought together via a "higher order meaning-making device" (Goldman and Kane, 2014). A level of relatedness is inferred by the proximity of both the individual statements to each other and the proximity of the clusters to each other. Those located at a distance from each other are conceptualised by the participants as being less similar

than those that are neighbours.

Discretion and knowledge of the system is used by the analysts to propose the final number of clusters, each of which represents a concept within the system. The first output of the cluster analysis maps every single statement as an individual concept, in this case 47 individual concepts. The method then combines statements in turn according to similarity. In the first instance the

two statements that were most frequently combined by participants in phase 2 (Figure 1) form the first cluster. The analysis





continues stepwise combining individual statements to groups (cluster groupings) of statements, until the final solution places all statements in a single cluster. The challenge is then to decide which arrangement of cluster groupings best represent the concepts and system (Kane and Trochim, 2007). The final solution depends on the context in which the map is to be used and level of detail required. It is helpful to sort in two stages, an initial analysis that follows Wards cluster analysis through to its

completion of one single cluster. Following this a second detailed review begins with agreeing a minimum and maximum number of concepts (Figure 3b) to represent the system. Beginning with the minimum number of clusters, the statement, or cluster that combine at each step are examined closely looking for a sense of cohesion, mutual understanding and participant interpretation within the grouping and a difference between the two newly formed concepts. This process is repeated until the clusters are recognised by the participants as concepts that represent the system under scrutiny.


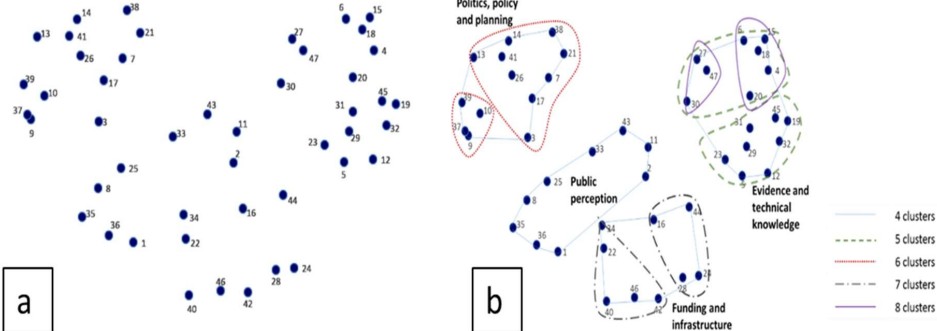

**Figure 3** a. multidimensional scaling showing similarity and dissimilarity of the 47 barrier statements b. Ward's hierarchical cluster analysis, clusters 4-8

### 3.2.3 Phase 3 Participant interpretation and analysis

The final step of the method brings participants together to interrogate and reflect on the analysis and interpretation of the previous two phases (Bickman et al., 2016; Newing et al., 2011). On the 30th April 2018 in Warrington, North West England, 19 practitioners attended a workshop to contribute to phase 3 of the GCM with Ketso methodology. The participants were recruited through snowball sampling (Peters, 2017). The 24 participants of phase 2 were invited and asked to recruit further participants from their own catchment partnership and flood risk management professional networks to widen the participation

of practitioners to interrogate the findings of the study more widely. The aim of the workshop was twofold, first to share and consult on the mapped NFM delivery system according to the ways in which the practitioners approach and organise their knowledge production (Wietske et al., 2009) and second to interrogate the perception of the importance of individual barriers. Two separate working frameworks can be assigned to the two practitioner groups: flood risk management (Johnson and Priest, 2008) and integrated catchment management (Falkenmark, 2004). The statement lists, their ranking and the grouping of the





barrier statements into concepts and their ranking were discussed and reviewed by referring to these working frameworks as a
       catalyst to consider the socio-ecological-technical system as a driver for differences in perceptions and experiences.

## 4.    Results

       The 253 barriers generated at phase 1 were sorted and reduced to the final 47 (See Appendix1). The cluster map presented at
the workshop consisted of seven concepts (Figure 3b): *politics, policy and planning, public perception, funding, infrastructure,
       evidence* and *technical knowledge*. Group labels were selected in discussion with practitioners during the feedback and
       interpretation workshop aided by the labels generated through the sorting and ranking exercise.

       The placement and appearance of concepts on the map indicates a degree of relatedness (Figure 3a). Those concepts that are
mapped in close proximity to each other are statements often grouped together by participants, revealing that they are
       conceptually similar, for example politics (Figure 3b). Those mapped over a larger areas, or are more disperse, signal they are
       conceptually more heterogeneous, for example public perception (Figure 3b).

       An exploration of values and perceptions of the seven concepts that form the delivery system are made through ranking data,
with a mean score calculated for each concept grouping (Figure 4). The individual barriers that fall within each concept (see
       S1) were given a score according to the perception of importance. If ranked in the top 5 by participants it received a score of
       6, if placed between 5-10 a score of 5, between 11-20 a score of 4, 21- 30 a score of 3, 31-40 a score of 2 and the bottom 7
       barriers received a score of 1. From this a mean barrier score for each concept is calculated (Figure 4.)

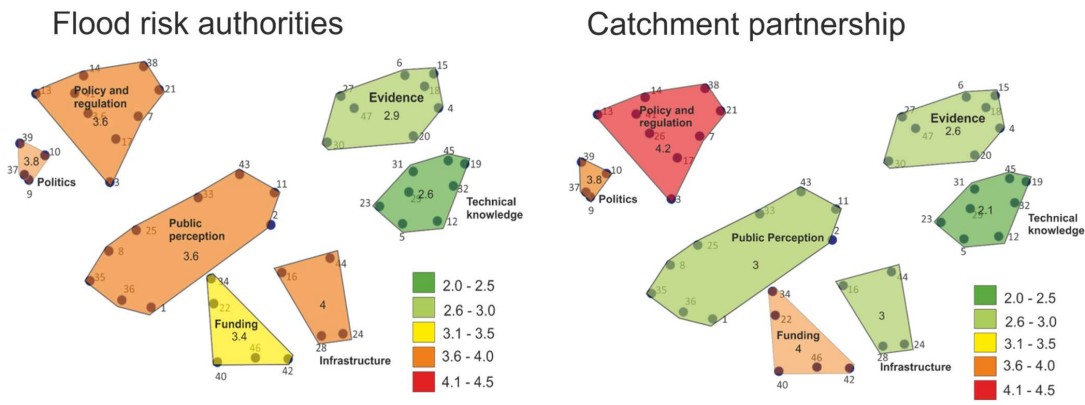






Figure 4. Comparison of flood risk authorities and catchment partnership and the perceived importance of the grouped NFM barrier concepts. Mean rank importance score: Top 5 barrier = 6, 5-10 =5, 11-20 = 4, 21-30 = 3, 31-40 = 2 and bottom 7 barriers = 1

Both practitioner groups ranked the *evidence and technical knowledge* barrier concepts as being of lower importance as a barrier to delivery, compared to socio-organisational barriers of *policy and regulation*, *politics* and *funding* (Figure 4).

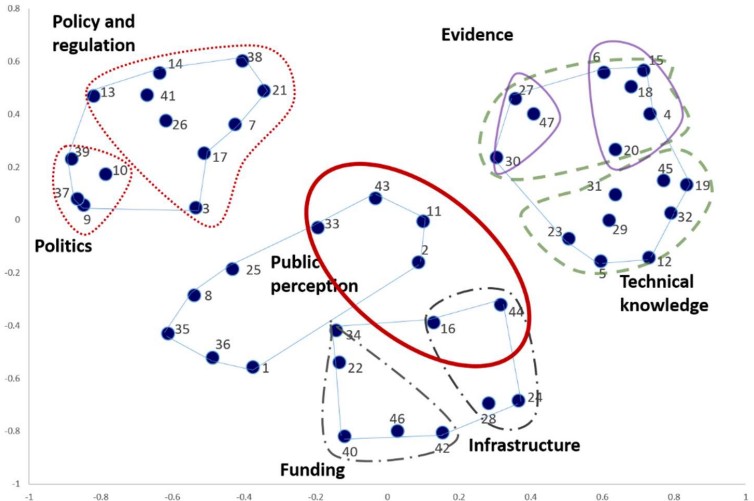

Figure 5. Bridging statements reveal barriers that require an integration of concepts

The placement of statements onto the map results from either, participant's perception of similarity, and indicate participants have regularly sorted them into the same groups in phase 2 or by their difference, in that they are rarely grouped together in phase 2, in such cases multidimensional scaling places them on the map at a distance. However, when there are differences in understanding, perception, or interpretation of meaning by practitioner groups the perceived difference can result in a statement regularly sorted by one group in to a particular concept cluster, whilst another group of actors recognises it as belonging
elsewhere. In this situation these statements are described as bridging different concepts. The barrier statements highlighted with the red oval (Figure 5) within the *public perception* and *infrastructure* concept were thought to be a result of bridging different concept clusters: 33 (The long term management and maintenance responsibilities are not clear), 43 (The timescales required for many NFM processes are long and do not match with our expectations of instant results), 11 (Failure - who would be liable? The first failure will be disproportionately scrutinised), 2 (Space - The interventions require a lot of space and the
UK has a growing population), 16 (There is still a reluctance to use 'new techniques' amongst the community this includes the



public, farmers, consultants and planners) and 44 (There is not a general acceptance from funders that monitoring is needed). The participant interpretation and analysis workshop deemed that the bridging statements were an important concept and should be recognised within the concept map. The barriers were therefore grouped together as a new concept and renamed as *cross-cutting issues*.  This led to a redrawing of the concept maps (Figure 6), including the loss of the concept, infrastructure

with two of its barriers (16 and 44) moving to the new *cross-cutting issues* concept grouping. The remaining two *infrastructure* statements (24 and 28) were moved into the *funding* concept, they refer to flood defence payments and cost benefit analysis respectively, therefore a strong argument can be made for their placement within a *funding* group. Public perception became a smaller group of 5 barrier statements plotted closely on the map, showing conceptual similarity of the grouping. However one further barrier was moved from *funding* concept, as the workshop participants thought it would be better placed in the

*public perception* concept, supported by its close proximity to the public perception cluster (Figure 5), this was barrier 34 (The public are not open to NFM techniques (leading to nimbyism), they resist change and feel safer with familiar options like dredging and have a poor understanding of risk and uncertainty). The changes were made as our methodological research approach is driven by knowledge co-production, including interpretation of results, the practitioners were of the opinion that these distinctions were useful for the wider transdisciplinary NFM study and therefore barrier number 34 was moved to the

new *public perception* cluster. The supplementary data gives the details of each individual barrier statement and the cluster which it was placed in following phase 3 of the GCM with Ketso methodology.





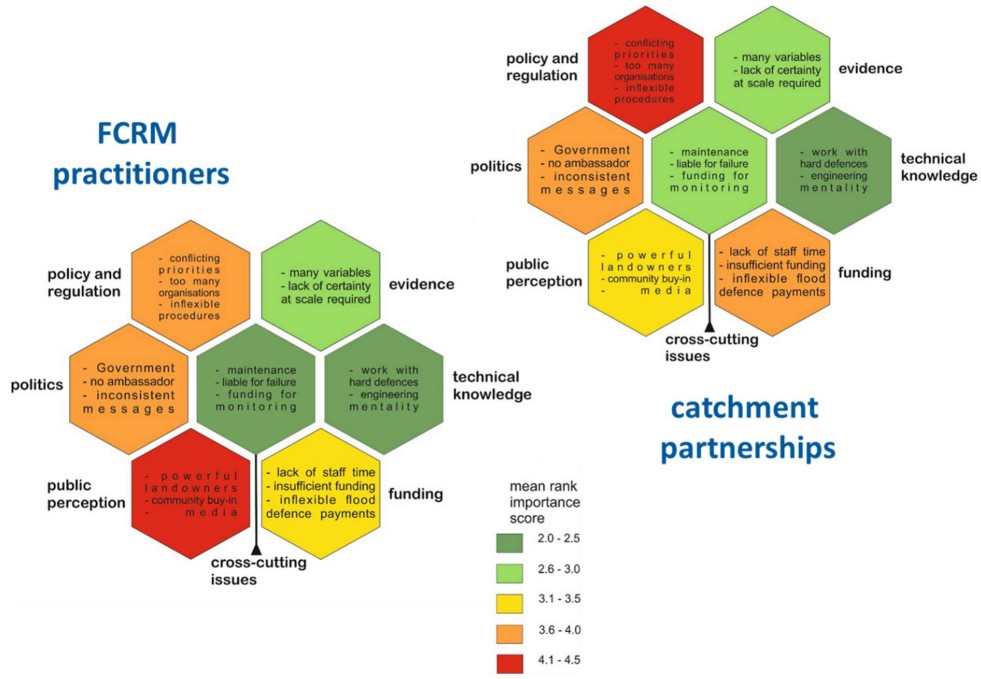

Figure 6. The revised concept maps following phase 3 participant interpretation and analysis

The method revealed several individual barrier statements with large differences in perceived importance between practitioners contributing to catchment partnerships and FCRM practitioners. Table 2 details the five with the largest differences.

**Table 2** Barrier statements from group concept mapping method showing large difference in perception between practitioner groups.

| | | FCRM | | Catchment Partnership | |
|---|---|---|---|---|---|
| | | Mean importance | rank | Mean importance | rank |
| 13 | There are too many agencies, leading to poor communication, lack of coordination, inconsistencies and/or a lack of understanding | 3.3 | 44 | 5.5 | 8-9 |
| 26 | It is difficult to work in partnership with so many organisations. There is not enough knowledge sharing, there is competition and there are differences in priorities | 5.15 | 20 | 5.5 | 8-9 |
| 25 | The time to develop a collective community buy in is significant | 5.7 | 4 | 4.75 | 30-32 |


| 22 | Organisations do not resource staff to do these types of projects; They can be on short term contracts so there is no consistency or they do not have the capacity or time | 4.5 | 27 | 5.9 | 4 |
| 24 | Flood defence payments are based on reducing flood risk to households, It is difficult to demonstrate NFM contribution to this | 5.6 | 5-6 | 4.7 | 33-34 |


## 5. Discussion

### 5.1 Public perception

FCRM practitioners ranked individual barrier statements in the *public perception* grouping highly (S1); time (4), landowner power (5-6) and a resistance to change (7-10), the media (11-15) and how a landscape should look (11-15). Catchment

partnerships have a distinctly different attitude to public perception as no individual *public perception* barrier ranked in the top ten, most received a middle ranking (see Appendix 1). FCRM works with communities at risk of flooding, anecdotal evidence collated from flood affected communities by Platt (2019), highlights a widespread public suspicion of authorities and a common belief their particular flood event was contributed to, if not caused by, human error and the mismanagement of flood structures upstream. Flood risk management has been described as a blame game (Cowen and Delmotte, 2019; Krieger, 2013),

where regulatory authorities must negotiate public scrutiny and accountability, further complicated by an ongoing debate as to whether responsibility for reducing flood risk lies with government, organisations managing flooding or with society as a whole (Butler and Pidgeon, 2011; Klein et al., 2017). Additionally a growing body of literature links social inequalities and living in flood risk zones (Fielding, 2018; Walker and Burningham, 2011) and it is against this backdrop that FCRM or any other organisation interested in expanding NFM to a mainstream flood management strategy must work; however, this context

for delivery of NFM was not identified as a barrier by catchment partnerships. Whilst it is not yet clear whether catchment partnerships ranking of *public perception* as a barrier of lower importance is a result of their skill in public engagement, or from a lack of knowledge and exposure to the complexities of working with communities living with flood risk, or driven by an alternative factor related to the conceptual heterogeneity of the grouping. The workshop discussion did reveal that catchment partnerships tended to see a reluctance to use techniques (statement 16) and long-time scales for NFM processes (statement

43) as a barrier caused by unwilling or uninformed practitioners rather than the public. The GCM concept maps support this finding as catchment partnership practitioners' placed these barrier statements within the *technical knowledge* concept, compared with FCRM practitioners who placed these barriers within *public perception*. A finding that suggests that FCRM practitioners do not perceive that they have agency to promote mainstream adoption, power lies with the public who are not supportive. Public perception and the disparity in its perceived importance to mainstream NFM delivery is an area for further

research.

### 5.2 Funding



The *funding* barrier cluster is ranked as having a greater importance as a barrier to NFM delivery by catchment partnerships than FCRM practitioners (Figure 6), however when individual barriers are examined, this broad distinction screens a more nuanced picture of funding and NFM delivery. For FCRM practitioners the difficulty of demonstrating a reduction of flood risk to households is ranked (5-6), for catchment partnership this barrier is of much lower importance (33-34) (Table 2). Catchment partnerships instead place, staff resources (4), availability of funding (3) and difficulty in applying cost-benefit analysis (10-12) as barriers of much greater significance. These differences illustrate that the two practitioner groups are working within different funding systems from which to finance NFM delivery.

FCRM, resources are principally derived from public funding with stringent conditions attached (DEFRA, 2011), one of which is to reduce flood risk to 300,000 homes by 2021. If criteria to assign central government flood funding, reduction of flood risk to households, is not met by NFM, FCRM are unable to progress and support delivery as a mainstream flood management option. Host organisations of catchment partnerships are charitable bodies, the narrative is that this enables access to a wider number of funding sources. This flexibility makes them suitable to coordinate activities of a range of actors who access and contribute funds and resources to deliver activities of mutual benefit and according to local need (Bide and Cranston, n.d.; Environment Agency, 2017). This study does not allow for an analysis of whether this is the experience in practice, however the results suggest that catchment partnerships are experiencing difficulties in securing financial resources for NFM delivery. Without a dedicated resource stream it is probable that NFM projects will be developed and delivered in an unsystematic ad-hoc way, strongly influenced by criteria set by the sources of competitive project funding. It is worth noting that catchment partnership practitioners do not rank a demonstration of a reduction of flood risk to households as important, indicating that there is a gap in their experience and/or understanding of conditions attached to central government flood funding.

### 5.3 Politics and Policy and regulation

The most important barrier to the adoption of catchment wide NFM according to both groups of practitioners is barrier 37, which describes a lack of commitment from government, particularly in conflicting and changeable messaging. The interpretation workshop (phase 3) gave an opportunity to explore this result further. To support this position practitioners referred to a number of recent experiences to justify their perception that government did not actively support the adoption of catchment wide NFM, including; technical evidence ignored in the face of pressure from land owners in Somerset (Cowen and Delmotte, 2019), reversal on introducing statutory requirements for sustainable drainage (Wingfield et al., 2019) and that flood management is only a topic of concern immediately following a flood, 'what we need is a good flood to get the attention of decision makers'. However, practitioners also reported that the situation was in a state of change. Policy guidance encouraged the adoption of NFM, a single one-off NFM project fund of £15 million had been recently allocated (Webb et al., 2018) and an NFM evidence directory (Burgess-Gamble et al., 2017) was due to be released. Practitioner feedback demonstrates a confused picture of government commitment to NFM adoption. Studies into the role of government in climate change adaptation have pointed to a failure of coordination between levels of government, from national to local, particularly with a lack of resources at the local level (Biesbroek et al., 2013), whether this applies to NFM delivery is an area for further analysis.




The policy and regulation concept displays the greatest divergence in opinions between the two practitioner groups. The most striking, is barrier 13, "There are too many agencies leading to poor communication, lack of coordination, inconsistencies and/or a lack of understanding". The catchment partnerships ranked this barrier 8-9 whereas FCRM ranked it at 44 (Table 2). This represents a large difference in perception of importance. A similar idea to barrier 13 is captured in barrier 26 "It is difficult to work in partnership with so many organisations. There is not enough knowledge sharing, there is competition and there are differences in priorities" ranked 20 by FCRM practitioners and 8-9 by catchment partnerships. The difference in perception is not as great as barrier 13, but again greater importance is placed on this barrier by practitioners from catchment partnerships.  From the GCM methodology it is not possible to say whether this is a result of an opinion from FCRM practitioners that they work well with other organisations or whether the ranking is driven by the opposite position that cross sectoral working is not important. Given that the Environment Agency's own internal review (Table 1) conducted in 2019 concluded that a culture change within flood risk management is required to open up collaborative working, suggests that perhaps the latter is most probable. Instead of approaching NFM delivery from a catchment based perspective, FCRM practitioners are attempting to fit NFM delivery within the traditional engineering led approach, unaware or resistant to the integrated nature of NFM delivery, and do not agree that other agencies are needed to adopt NFM as part of a mainstream flood management strategy. Catchment partnerships on the other hand were established to bring together interests of different land and water management bodies (Cascade, 2013), as such they would be expected to have a good understanding of whether different sectors and organisations are well aligned to facilitate delivery and greater collaborative working required to support NFM delivery. The high ranking of barrier 13 and 26 suggests they value collaborative working, but it is not operating effectively, whether this is unique or particular to the NFM delivery system or other activities and governance of catchment partnerships more broadly is an area for further research.

**5.4 Evidence and technical knowledge**

Both flood risk authorities and catchment partnerships recognise that because of the large number of physical processes involved in NFM there are scientific uncertainties in their optimal use. However barriers within the *technical knowledge* concept notably have not received a high ranking by either practitioner groups (within the top twenty - S1). Conversely, during the interpretation workshop practitioners, in particular those from FCRM did not agree that the *evidence and technical knowledge* concept group did not act as an important barrier to delivery. There was an insistence that it should be understood as one of the most important barriers. To support this perception practitioners described a technocratic linear process of first filling evidence gaps, leading to policy change which then opens up funding streams. This position was somewhat contradicted when the workshop went on to discuss the number of supportive policy documents that have been in circulation for over a decade. This discrepancy of whether there was sufficient evidence to formulate policy or whether there was a failure in policy delivery could be explained in part by the practitioners who engaged with the project. They are technical experts who are responsible for delivering land and water management interventions conceptualising that task through a natural sciences



outlook. If their rating of the importance of individual barriers was influenced by familiarity, their expertise and comfort with accessing and using the technical evidence base are likely to lead to a lower ranking of those barriers compared to social or political barriers. The participants identified this in the interpretation workshop and called for greater inclusion of senior managers responsible for strategic planning and policy makers in any further research.

510

### 5.5 Cross-cutting issues

The group of cross-cutting issues contain barriers described by one participant at the interpretation workshop as the, "parts that don't belong to anyone and get easily forgotten". These barriers have been described as bridging statements, indicating linkages or interconnections between concepts. When phase 2 data were reviewed to see which concepts the statements were bridging, it was only possible to identify this pattern for two barrier statements: 16 (a reluctance to use new techniques) in which participants had grouped the barrier most frequently with both *public perception* barrier statements (34, 35 and 36) and a *technical knowledge* statement (5). Barrier 43 (long time scales for NFM processes) had been grouped in both *technical knowledge* (12 and 19) and *public perception* (25 and 34). Within the cross-cutting group these are also only barriers that were ranked highly (see Appendix 1). The remaining barrier statements in the cross-cutting group, 2, 11, 33 and 44 did not show any strong association with any of the other barriers. Why these particular barriers were not linked to other barriers within the identified 47 is unclear. It could be that their meaning was considered ambiguous or, represent components of the delivery system that this group of practitioners, were unfamiliar with, which would also explain their low importance ranking.

### 6. Lessons from applying the approach in practice

The GCM with Ketso method worked well, outputs combined with the interpretation workshop have allowed an exploration examining interconnections and perspectives of practitioner groups, identifying problems for further investigation. The group concept mapping method allowed for a relatively large number of practitioners to input for this type of knowledge co-production study – 89 individuals across three phases. However, the portion of the study that produced the basic concept maps was limited to the opinions of 24 individuals. The conceptual framework dividing the NFM delivery system into seven areas should be tested in practice and reviewed against published theories. A number of findings, particularly that the greatest constraints to delivery come from socio-organisational factors support the conclusions of studies examining barriers to the implementation of linked environmental management policies, such as climate change adaptation (Biesbroek et al., 2013; Moser & Ekstrom, 2010) and blue green infrastructure (O'Donnell et al., 2017; Thorne et al., 2018) and it is thought likely that the seven concepts underpinning NFM delivery are translatable nationally and internationally.

The interpretation workshop is a key component of the group concept mapping methodology, however it must be understood as a snapshot in time, in what is an evolving area of research and practice. A number of changes to the concept maps recommended in the workshop were implemented, while the alterations did not change outputs for individual barrier statements, it did result in changes to the mean rank importance score of *public perception*, identifying it as the greatest barrier





concept grouping to NFM delivery for FCRM practitioners. As discussed above the newly formed group called *cross cutting*
*issues* revealed uncertainty as to whether the bridging nature of the statements are driven by practitioner's lack of familiarity
or ambiguity in phrasing. Research on risk and uncertainty show there is a difference in perception within organisations from
practitioners with differing authority and responsibility (Höllermann & Evers, 2017). Inclusion of a greater number of senior
managers with experience of strategic planning and policy making would have enabled a comparison of the perspectives from
this actor group. In the wider transdisciplinary study there may be a need to revisit the problem identification phase with this
group of actors (Pohl et al., 2021).

## 7. Conclusion

The perceptions of barriers to NFM delivery between two actor groups, FCRM and catchment partnerships have been examined
and compared as a problem identification phase of a wider transdisciplinary study. Transdisciplinary studies begin with
problem framing and structuring perspectives of practitioner's and researchers across disciplines and sectors to develop mutual
understanding as basis for further research and cooperation. Topics for further analysis and key findings of the problem framing
phase are: public attitudes to NFM techniques, cross sector collaboration and partnership working, FCRM practitioner culture
and, what is a successful outcome? - Clarity from central and local government messaging.
There is broad agreement that the UK government is not sufficiently supportive and impeding delivery of NFM, whether this
is a problem of central government and national policy making, or associated with local government, or found across all
administration levels of government remains unclear. For FCRM practitioners, *public perception* was the most important
barrier to NFM and for catchment partnerships, *policy and regulation*. The findings of the group concept mapping methodology
support the literature on barriers to environmental management policy adoption more widely, that the greatest constraints to
change come from socio-organisational factors rather than a lack of technical knowledge, however it must be acknowledged
that practitioners queried this finding in the interpretation and analysis workshop. The implications of these findings are
significant, as the areas identified within this study of greatest concern are those that often receive the least research funding.
Current research efforts often focus on approaches and attempts to improve technical knowledge, this in part may reflect the
nature of research funding as operating within disciplinary boundaries, but for NFM to be successful further system based
research is needed to move beyond disciplinary boundaries and reflect the integrated nature of NFM delivery
The group concept mapping methodology has been employed as a problem identification method allowing for the active
participation of a multidisciplinary group of 89 practitioners, from which the system of NFM delivery has been revealed. This
holistic process has allowed for an interrogation of different perspectives between practitioner groups who have different
working conceptualisations and knowledge systems. Two separate working frameworks can be assigned to the two practitioner
groups: flood risk management (Johnson and Priest, 2008) and integrated catchment management (Falkenmark, 2004). Beyond
the context of UK NFM delivery, the approach described in this paper can be utilised across the globe to support the delivery
of NBS for flood management in providing a framework for systematic problem oriented research. NBS as a practice based





approach that explicitly includes a second pillar of social safeguarding within traditional conservation efforts (Cohen-Shacham et al., 2019; Ruangpan et al., 2020) requires an expansion in knowledge about how to implement them across scales, contexts and people. Concept maps create visual representations of the networks of ideas to describe how a group thinks about an issue

(Goldman and Kane, 2014; McLinden, 2013), allowing for problem oriented integration of different knowledge bases, across sectors and disciplinary framings. The maps expand the resulting knowledge beyond the context in which it was generated by encouraging concepts to be deconstructed, restructured or emerge (Rosas, 2017).

An interpretation of the study is that the FCRM sector is resistant to change, a means of lessening isolation from other sectors can be achieved through social learning via knowledge co-production. In this respect the study approach has been particularly

impactful, participating practitioner's developed mutual understanding, constructed a map of concepts underpinning the NFM delivery system and explored differences in perceptions impeding delivery. This study is the first step towards identifying means of encouraging the adoption of NFM as a mainstream flood management technique using a transdisciplinary approach of problem framing with NFM delivery practitioners.


**Appendix 1**

Appendix 1. 47 individual barrier statements in their seven concepts, mean importance and rank according to practitioner groups. The top ten barriers are underlined to highlight them. Those in bold typeface indicate a large difference in perceptions between the two actor groups






| | | List of barrier statements within each cluster | FCRM Mean importance | FCRM rank | C.P. Mean importance | C.P. rank |
|---|---|---|---|---|---|---|
| Politics | 9 | Knee jerk political reactions following flood events or changing political whims that the regulator has to respond to lead to inconsistencies | 5.4 | 11-15 | 5.4 | 10-12 |
| | 10 | There is no ambassador or clear authority of NFM. For example who should engage and educate landowners? | 4.5 | 26 | 5.2 | 18-22 |
| | 37 | The government. - The buy in from MPs' that is needed is not there. They give the public conflicting messages, for example both working with natural processes and dredging and they work to short government timescales | 5.9 | 1 | 6.3 | 1 |
| | 39 | Defra / NFU relationship | 3.9 | 37-38 | 3.9 | 40-41 |
| | 3 | Conflicting priorities for land; food, housing, biodiversity, flood defence, and conservation | 5.5 | 7-10 | 6.1 | 2 |
| Policy and regulation | 7 | Vision - there is no coherent vision of desired outcomes at the catchment scale. Too much talking not enough doing | 5.4 | 11-15 | 4.5 | 36 |
| | 13 | There are too many agencies, leading to poor communication, lack of coordination, inconsistencies and/or a lack of understanding | 3.3 | 44 | 5.5 | 8-9 |
| | 14 | Poor integration of planning and design policy on a catchment scale; strategically and politically | 5.5 | 7-10 | 5.8 | 5 |
| | 17 | Locked in - standardised approaches and procedures mean regulators are locked in and are inflexible, for example it's difficult to deliver projects with multiple benefits | 4.3 | 28-29 | 5.6 | 6-7 |
| | 21 | Current organisation (or lack of) means we are in danger of a scatter-gun approach | 5.3 | 16-17 | 4.9 | 27-28 |
| | 26 | It is difficult to work in partnership with so many organisations. There is not enough knowledge sharing, there is competition and there are differences in priorities | 5.15 | 20 | 5.5 | 8-9 |
| | 38 | Some regulatory procedures make adoption difficult e.g. the reservoirs act, countryside stewardship agreements, the common agricultural policy | 5.2 | 18-19 | 5.4 | 10-12 |
| Public perception | 41 | Bureaucracy (planning, licensing etc.) | 4.2 | 30-31 | 4.75 | 30-32 |
| | 1 | Landowners have power and influence and are not convinced by NFM. They will only agree if they are paid compensation | 5.6 | 5-6 | 5.25 | 14-17 |
| | 8 | Stakeholders who do not want to be involved have the power to put a stop to projects | 4.2 | 30-31 | 5.2 | 18-22 |
| | 25 | The time to develop a collective community buy in is significant | 5.7 | 4 | 4.75 | 30-32 |
| | 34 | The public are not open to NFM techniques (leading to nimbyism), they resist change and feel safer with familiar options like dredging and have a poor understanding of risk and uncertainty | 5.5 | 7-10 | 5.1 | 23-25 |
| | 35 | The media portrayal of flood management | 5.4 | 11-15 | 5.2 | 18-22 |
| | 36 | The perception of how a landscape should look are ingrained e.g. Cumbrian sheep grazed hill sides and managed rivers. This makes changes difficult; politically, economically and culturally | 5.4 | 11-15 | 5.25 | 14-17 |


| | | FCRM | | C.P. | |
|---|---|---|---|---|---|
| | List of barrier statements within each cluster | Mean importance | rank | Mean importance | rank |
| **Funding** | | | | | |
| 22 | Organisations do not resource staff to do these types of projects; They can be on short term contracts so there is no consistency or they do not have the capacity or time | 4.5 | 27 | **5.9** | 4 |
| 24 | Flood defence payments are based on reducing flood risk to households, It is difficult to demonstrate NFM contribution to this | 5.6 | 5-6 | 4.7 | 33-34 |
| 28 | Option selection is currently based on cost benefit analysis. The benefits and costs of NFM are not understood so it is excluded | 4.9 | 23-25 | 5.4 | 10-12 |
| 40 | Funding - Insufficient, difficult to access and inappropriate; for example will not pay for staff time or does not join with other funding streams | 5.3 | 16-17 | 6 | 3 |
| 42 | The efficiency culture is to reduce spend not to maximise outcomes and NFM is not always the least cost option | 4.1 | 35-36 | 4.7 | 33-34 |
| 46 | Schemes will not be paid for if there is not an immediately adjacent community at risk | 4.3 | 28-29 | 5.1 | 23-25 |
| **Evidence** | | | | | |
| 4 | Uncertainties - Will there be unintended consequences? Could the situation be made worse e.g. more trees causing blockages? Can we stop a natural process? | 4.15 | 32-34 | 5 | 26 |
| 6 | Evidence - there is not enough evidence | 5.1 | 21 | 4.7 | 30-32 |
| 15 | A lack of certainty of which measures work in which conditions, particularly on a large scale when urban areas are downstream | 5.5 | 7-10 | 5.2 | 14-17 |
| 18 | Scientific uncertainties - There are many variables involved and to understand: changing rainfall patterns, cumulative impacts, influence of scale, climate change and the behaviour of the floodplains | 5.8 | 3 | 5.6 | 6-7 |
| 20 | Trial plots are small scale and have not taken into account mixed land use | 4.15 | 32-34 | 3.8 | 42 |
| 27 | There is poor knowledge management of past projects | 4.15 | 32-34 | 4.9 | 27-28 |
| 30 | Practitioners don't have access to visualisation and mapping tools or data | 3.7 | 43 | 4.4 | 37-38 |
| 47 | Incorrect application of best practice leading to a lack of innovation or pragmatism | 3.9 | 37-38 | 3.9 | 40-41 |
| **Technical knowledge** | | | | | |
| 5 | Engineering mentality - Flood defences built with concrete offer certainty, the only solutions ever considered are engineering ones. There is a lack of sustainability training amongst engineers | 5.0 | 22 | 5.1 | 23-25 |
| 12 | Solutions are believed to come from technology, we expect to be able to control our environment | 3.85 | 39-40 | 3.5 | 44 |
| 19 | NFM is only suitable for low order events | 3.00 | 45-46 | 3.2 | 45-46 |
| 23 | Academic training and courses don't cover NFM | 3.7 | 41 | 4.4 | 37-38 |
| 29 | It's not the full solution and needs to often go hand in hand with harder defences | 5.2 | 18-19 | 4.8 | 29 |
| 31 | Beavers are not seen as NFM | 3.0 | 45-46 | 1.9 | 47 |

| | | FCRM | | C.P. | |
|---|---|---|---|---|---|
| | List of barrier statements within each cluster | Mean importance | rank | Mean importance | rank |
| **Technical knowledge** | | | | | |
| 32 | Modelling - It is very difficult as whole integrated catchment processes need to be modelled. There is no single accepted technique or measure of success. It is expensive and difficult to fund | 4.9 | 23-25 | 5.2 | 18-22 |
| 45 | Not reliable enough a solution for public protection | 2.3 | 47 | 4.6 | 35 |
| **Cross-cutting issues** | | | | | |
| 2 | Space - The interventions require a lot of space and the UK has a growing population | 6.6 | 42 | 3.6 | 43 |
| 11 | Failure - who would be liable? The first failure will be disproportionately scrutinised | 4.1 | 35-36 | 3.2 | 45-46 |
| 16 | There is still a reluctance to use 'new techniques' amongst the community. This includes the public, farmers, consultants and planners | 5.85 | 2 | 5.2 5 | 14-17 |
| 33 | The long term management and maintenance responsibilities are not clear | 4.9 | 23-25 | 5.3 | 13 |
| 43 | The timescales required for many NFM processes are long and do not match with our expectations of instant results | 5.4 | 11-15 | 5.2 | 18-22 |
| 44 | There is not a general acceptance from funders that monitoring is needed | 3.85 | 39-40 | 4.3 | 39 |

**Author contribution**

**Thea Wingfield** – Conceptualisation, methodology, formal analysis, investigation, data curation, writing original draft and visulaization

**Neil Macdonald** — Conceptualisation, validation, writing – review and editing, supervision, funding acquisition

**Kim Peters** — Validation, writing – review and editing, supervision





**Jack Spees** – Funding acquisition, supervision, resources, writing – review and editing
The authors declare that they have no conflict of interest

**Acknowledgments:** We would like to thank all of the practitioner research collaborators who took part in this study. A special thanks goes to Ribble Rivers Trust, River Restoration Centre and the Environment Agency for assisting in recruitment,
organisation and hosting research events.
**Funding:** This work was supported by Natural Environment Research Council, Grant Award Number NE/L002469/1

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
