# Peer review of "Barriers to mainstream adoption of catchment wide Natural Flood Management, a transdisciplinary problem framing study of delivery practice"

_Hydrology and Earth System Sciences, 2021_

## Author Response (AR1)

**School of Environmental Sciences, Geography**

Roxby Building
Chatham Street
Liverpool
L69 7ZT

T  0151 794 2510
F  0151 793 2866
E  tajw@liverpool.ac.uk

www.liv.ac.uk/geography

**RE:** hess-2021-404

20 October 2021

Dear Thomas Thaler,

We are pleased to resubmit for publication the revised version of manuscript hess-2021-404 "*Barriers to mainstream adoption of catchment wide Natural Flood Management, a transdisciplinary problem framing study of delivery practice*". We highly appreciate the positive feedback and constructive guidance from the three reviewers. We would like to thank all three reviewers and the helpful and insightful comments raised.

Please find enclosed with this letter our revised (marked-up and clean) manuscript and below a table detailing our responses point-by-point to the reviewer comments.

Yours sincerely

Dr Thea Wingfield. On behalf of Jack Spees and Professors Neil Macdonald, Kimberley Peters

|  | comment | Response |
|---|---|---|
| RC1 | Overall, I really enjoyed reading this paper - well done!

As the paper investigates mainstreaming natural flood managment, the criticism of NFM should also be mentioned within the introduction (e.g. that it would be overwhelmed with extreme floods).

The Ketso method and its applications is well described which makes the method-chapter very transparent. But sometimes it gets lost in details, which makes some texts hard to read. | Thank you for your positive comments.

We are pleased to hear you enjoyed reading the paper and appreciate your helpful suggestions.

We have updated the introduction to mention the limitations/criticisms of NFM that during very large storm events the NFM interventions may be overwhelmed. (line 43)
- *Whilst some such as Dadson et al. (2017) have raised concerns that during extreme floods measures could be overwhelmed, others such as* |

A member of the
Russell Group

| | | | |
|---|---|---|---|
| | Maybe you can summarise some parts or move them to the appendix.

I would appreciate if you would clearly highlight your most important statements within the conclusions. | *Norbury et al., (2021) have observed that this may not necessarily be the case, subject to design and magnitude of event. . Rather the paradigm shift for flood management in adopting NBS within their programs is a contribution to flood risk reduction alongside greater environmental and social goods (Connelly et al., 2020; Fenner, 2017; Hanson et al., 2020)*

We have reviewed the section describing the Ketso methodology with the view of improving clarity by making it more succinct. In light of reviewer 2 highlighting that they liked the "precision" and and reviewer 3 commending "practical insights for structuring discussions on mainstreaming issues" we have taken the decision not to move parts of the method chapter to an appendix. But we have found places to improve clarity:

(line 247) *GCM with Ketso is formed of six steps, which for the purposes of this transdisciplinary problem framing study were grouped into 3 phases; phase 1 qualitative statement generation, phase 2 quantitative statement sorting and ranking and phase 3 interpretation (Figure 1).*
(line 296) *Phase 2 was not undertaken in a workshop setting, qualitative statement sorting and ranking (phase 2, Figure 1) was undertaken by 12 flood risk management professionals and 12 practitioners who contribute to catchment partnership either alone or in a small number of cases in pairs and groups of threes. The principal researcher provided guidance to the participants.*

(line 299) *Whilst national experts were the target of phase 1, statement generation; practitioners who work within North West England were selected for the second phase – statement sorting and ranking.*

(line 307) *Phase 2 involved each participant sorting the statements into groups that they felt contained similar or related ideas and classify the group by giving it a name.*
We will ensure that the conclusion highlights the most important barrier statements |
| RC2 | First of all, I need to admit that I really enjoyed reading the paper. I positively evaluate the construction of the field research and | We are very pleased that you enjoyed the paper and have praised the quality of the work. | |

| | especially how precisely the work was done and presented. This is not usual for the stakeholder evidence-based papers. I also think that paper is easy to read and that the main findings are highlighted. Also, the paper is well connected to existing NBS/NFM debates and devotes the part for terminological problem.

I missed one aspect that could be added in introductory/review parts before the own methodology is introduced: That is the one of self-engagement of land owners in NBS implementation and how this (rare) self-engagement is viewed (criticized) by FRM experts. Under the pressure of climate change, some land-owners are willing to restore historically changed hydrological conditions of their land, often at their own cost or giving up (agricultural) subsidies for missed production. This engagement shall be cherished and scaling-up potential shall be investigated (as this behavior moves out one of the key barriers – land-owner resistance). But what about coordination problem? I do not insist this debate must be incorporated in the paper, but if authors are open to do so, they might start with: https://link.springer.com/chapter/10.1007/978-3-030-23842-1_6 and follow with responses on this piece: https://link.springer.com/chapter/10.1007/978-3-030-23842-1_7 and https://link.springer.com/chapter/10.1007/978-3-030-23842-1_8.

Overall, I fully support the paper for the publication and congratulate the authors for the great work. | Thank you for your suggestion in including motivations of land owners even at their own financial cost. This is an important topic and was raised as a barrier to adoption within this study. However after reviewing the paper we felt that we could not give a sufficiently informed and nuanced discussion of landownership and NBS delivery. We felt however that it should be recognised and we have signposted within the paper as (line 83) *For example NBS for flood management requires land, and as such, land ownership, motivations of land owners, (Slavíková and Raška, 2019) balancing public and private interests and whether interconnected policy, legal and economic systems are supporting or hindering the engagement of land owners (Hartmann et al., 2019) and therefore the commitment of land is critical to mainstreaming NBS* |
|---|---|---|
| RC3 | This was a really nice paper to read. It illustrates how to engage different stakeholders in the framing of NFM in the UK through the use of group concept mapping and Ketso methods. The study addresses a very important topic of mainstreaming. It offers practical insights for structuring discussions on mainstreaming issues in NBS and other management and/or domains | Thank you for your positive comments we appreciate your considered response and support of the paper.

We have improved the clarity of the selection of participants for the different phases of the GCM with Ketso methodology |

also. In general, the manuscript is well structured and clearly written. Below are a few comments and questions:

Methods:

The methodological choices can be more explicit in a number of areas (and may be included also in fig. 1). For example:

Section 3.1 describes the participant identification more broadly in terms of the two practitioner groups. What was the criteria for selecting participants from these two groups in each phase? It would help to describe the participants/workshops in section 3.2.1 onwards by referring to the two practitioner groups they belong to. For example, was workshop 1 with flood risk authorities and workshop 2 with catchment partnership members? Also, in some phases the groups are mixed? Were the participants in the three phases the same? The numbers differ over time but were they (a subset of) the same participants from the 1st phase? Similarly, phase 3 uses snowball sampling techniques to recruit other participants- please explain this design choice and what added value it had in phase 3? I also assume the initial statements generated from both practitioner groups were combined and then qualitatively sorted and ranked separately by the flood risk authorities and catchment partnerships? Is this correct? If so, please clarify this somewhere in the text also.

Practitioner groups:

Section 3.1.2 (pg 8). Some additional details about the catchment partnerships would be helpful. Is there is a single catchment partnership "network" that exists in the UK with regards to integrated water management or are they specific to different regions/watersheds etc.? This is somewhat unclear from the text. Some examples of the types of actors that are part of these catchment partnerships, aside from the host NGO organization would also be helpful.
* * *
(line 254) *Phase 1 was completed in two sessions to capture the input from a broad group of water-focused practioners'. The first through a workshop at the River Restoration Conference in Blackpool on the 27th April 2016. 39 of the conference attendees took part in the workshop enabling access to a heterogeneous practitioner group drawn from across the UK, many of whom are expert in their fields including contractors, engineers, consultants, academics, environmental NGOs and government agencies (Wingfield, 2016). The second workshop was attended by 12 practitioners from the Environment Agency National Capital Programme Management Service (NCPMS), individuals responsible for delivering the Flood Risk Capital Programme. This team was selected as having the most comprehensive knowledge of flood risk management scheme delivery via their responsibility for managing the multi-million pound budget allocated to large flood management schemes across England.*
And
(line 296) *Phase 2 was not undertaken in a workshop setting, qualitative statement sorting and ranking (phase 2, Figure 1) was undertaken by 12 flood risk management professionals and 12 practitioners who contribute to catchment partnership either alone or in a small number of cases in pairs and groups of threes. The principal researcher provided guidance to the participants.*

Phase 3 used snowball sampling to *widen the participation of practitioners to interrogate the findings of the study more widely.* Our thinking behind our design choice was that as
*The aim of the workshop was twofold, first to share and consult on the mapped NFM delivery system..... and second to interrogate… perception*(s) and that we would be more successful in both aims by extending our network further and including practitioners who had not yet contributed.

Section 3.1.2 summarises information about catchment partnerships in a previous paper we undertake a review of the catchment partnership network and link it to water management frameworks, like integrated water management and flood risk management. We signpost readers to this paper but have avoided repeating the same

Results: I would like the authors to reflect on the following

1. Section 3.1.2 refers to the strategic potential of the catchment partnerships in leading and influencing NFM. At the same time, the authors recognize financial, organizational shortcomings. Does this research reveal new insights on the role of catchment partnerships in mainstreaming NFM?
2. What is the role of the group concept mapping technique in the mainstreaming problem. Is it a useful first step in identifying barriers from different perspectives or, does it also offer insights for exploring how the practitioner groups could identify solutions (based on their own strengths or resources of example) to help overcome some of these barriers?
3. How does this method (group concept mapping) compare to other participatory research methods like for example Q methodology with regards to examining mainstreaming problems?

Overall, I think this paper is a very nice addition to this special issue.

information (line 197) *A previous study identified that within the UK environmental governance system, catchment partnerships are well placed to co-ordinate delivery as the integrated water management framework that steers the movement is comparable and compatible to NFM in encouraging the delivery of multiple benefits coordinated at a catchment scale (Wingfield et al., 2019).*

Thank you for your useful reflections to consider in the results section.

Point 1 and 2 are related. In the introduction (line 109) the text states *The GCM method produces visual representations of what a group is thinking on a particular topic (Donnelly and Ph, 2016) and in doing so enables integrated problem identification, the primary component of transdisciplinary research (Jahn et al., 2012; Lang et al., 2012; Pohl et al., 2021).* AND (line 116) *our aim is to examine interdependencies and identify conceptual convergence within the delivery system. In doing so the study reveals conditions in which barriers to the delivery of NFM persist and begins to identify areas for further research and intervention points that could act as a catalyst for change (Eisenack et al., 2014).* In the results we are trying to draw out elements of a system through concept mapping (we have identified seven) to both consider these elements and the individual barriers from different perspectives and to identify intervention points. We discuss each of the seven elements in turn and whether it reveals new insights into NFM mainstreaming from the perspective of flood management or catchment partnerships For example (line 438) *The GCM concept maps support this finding as catchment partnership practitioners' placed these barrier statements within the technical knowledge concept, compared with FCRM practitioners who placed these barriers within public perception. A finding that suggests that FCRM practitioners do not perceive that they have agency to promote mainstream adoption, power lies with the public who are not supportive. Public perception and the disparity in its perceived importance to mainstream NFM delivery is an area for further research.*

| | | 3. Thank you for your interest in group concept mapping we have produced a methods paper that discusses GCM epistemology and discusses the approach within the umbrella of participatory approaches. In this case GCM has been employed within a transdisciplinary study for problem framing and bring together practitioners whilst minimising power imbalances and a framework to consider different perspectives. We felt there is sufficient material for a separate paper detailing the methodological approach and how it is well suited to problem-oriented integration and so have not expanded the discussion within this paper. |